# Foveated Downsampling Techniques

## Abstract

Foveation is an important part of human vision, and a number of deep networks have also used foveation. However, there have been few systematic comparisons between foveating and non-foveating deep networks, and between different variable-resolution downsampling methods. Here we define several such methods, and compare their performance on ImageNet recognition with a Densenet-121 network. The best variable-resolution method slightly outperforms uniform downsampling. Thus in our experiments, foveation does not substantially help or hinder object recognition in deep networks.

## 1   Introduction

The retinas of humans, monkeys, and many other animals have a high-resolution fovea. In humans, this disproportionate representation of the central visual field carries through the whole visual cortex, and eye movements to foveate task-relevant features are an essential part of vision. Deep convolutional networks are inspired by the primate visual system, but they usually lack foveation, which may be a limitation in some contexts. In humans, foveation allows both the wide field of view needed for tasks like visual navigation, and the high resolution needed for tasks like reading, without impractical brain size or metabolic cost. Similar benefits may await deep networks. Some previous studies have used a rough approximation of natural foveation, made up of several distinct images at different resolutions. In contrast, resolution changes gradually in natural systems. This may have benefits, but it is not clear how to arrange such a representation for input to a convolutional network. A circular image with high magnification at the centre wastes pixels at the corners. A polar representation does not, but it sacrifices translational equivariance. In summary, while foveation could potentially have benefits for deep networks, it is not clear when, or how best to implement foveation.

To help fill this gap, we compare several foveated downsampling approaches to uniform downsampling in object recognition. In this context, the different foveated methods perform fairly similarly to each other, and the best performs slightly better than uniform downsampling (top-1 validation accuracy 48.95% vs. 47.72%; Table 1). Therefore, foveation does not seem to be important for object recognition (which is unsurprising given the good performance of standard deep networks), but it does not greatly interfere either. This suggests that foveation could be incorporated into more general vision systems that perform multiple tasks, such as in robots that must recognize objects and also read text in the environment.

## 2   Methods

### 2.1   Network architecture and training

We trained deep networks on the ImageNet recognition task, with various kinds of downsampled images as input. In each case we used a DenseNet-121 [1] network with original hyperparameters

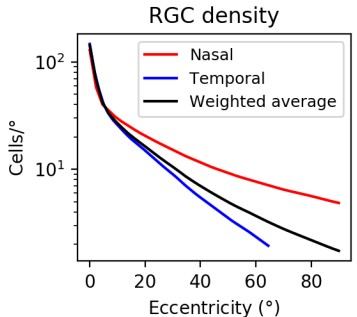

Figure 1: Estimate of retinal ganglion cell (RGC) density as a function of degrees from the fovea. We use estimates from [3], which provides data along the nasal-temporal axis. [4] shows that density is similar in temporal, dorsal, and ventral directions, but higher in the nasal direction. To calculate radially symmetric mean values, we sum nasal and temporal fits from [3] with weights 0.25 and 0.75 (to account for the fact that nasal density is atypical).

and training procedure, including random horizontal flips, batch size etc. We trained each network for 90 epochs, using SGD (initial learning rate 0.1, reduced by 10x every 30 epochs).

## 2.2   Downsampling techniques

*Uniform downsampling:* As a baseline method, ImageNet images were uniformly downsampled to a 32x32 resolution.

*Multi-resolution downsampling:* We produced a simple foveated representation composed of four $16 \times 16$ downsampled images with different magnifications. The first spanned the whole image, the second spanned the central half of the width and height of the image, the third a quarter the width and height, and the fourth an eighth. Several past papers have used a similar approach, e.g. [2].

*Polar retinal downsampling:* We sampled the image in polar coordinates, creating a rectangular image ($44 \times 23$ pixels) in which the long edge corresponded to the angle and the short edge the radial distance from the fovea. The density of samples in the radial direction declined with greater distance from the centre. We based the sampling density on retinal ganglion cell (RGC) density (see Figure 1). We used gaussian filters with radially increasing widths to reduce artefacts. See example in Figure 2.

*Cartesian retinal downsampling:* We sampled the image with the same radially-varying density as above, but created a circular image with strong barrel distortion (Figure 3), rather than a polar representation. This resulted in a transformation that better retains the translational equivariance property of convolutional networks, at the cost of wasting pixels in the corners.

## 2.3   Selection of image points to foveate

A saliency map was generated for each image with a DeepGaze II model [5]. This map estimated the likelihood of a human orienting to each pixel. Human gaze often orients to areas of interest such as faces and foreground objects, which often correspond to the target label. We selected the point of highest saliency, subject to a constraint that avoided points near image edges (as selecting a point near the edge would render much of the crop blank). Specifically, we only chose points around which at least 80% of a $256 \times 256$-pixel crop would fall within the image boundaries (Figure 4). If the resulting crop went outside the image boundaries, we extrapolated by copying edge pixels.

We sometimes chose multiple points in a single image. The highest-saliency points are typically close together, and contain similar information. To avoid selecting multiple similar points, we modified the saliency maps after each selection. Specifically, we subtracted a square-gaussian function from the saliency map, with a peak equal to the saliency at the chosen point, and a width of 60 pixels.

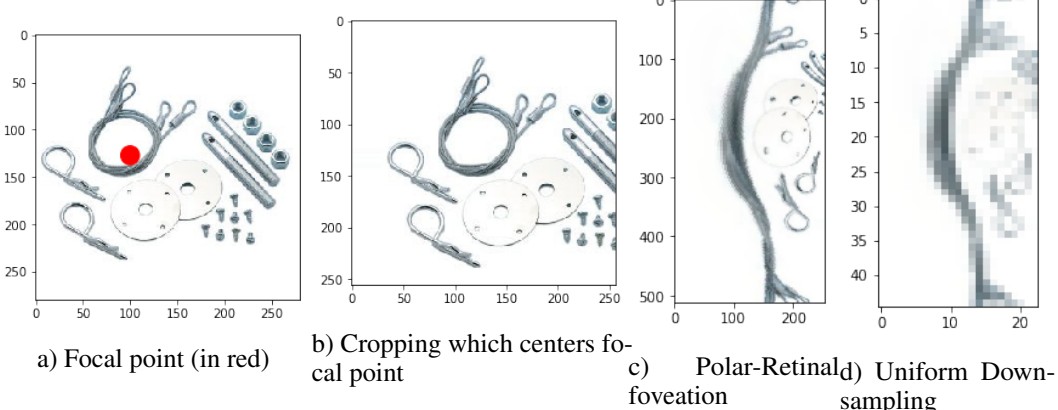

a) Focal point (in red)

b) Cropping which centers focal point

c) Polar-Retinal foveation

d) Uniform Downsampling

Figure 2: An example of polar-retinal downsampling. (a) The focal point (highest saliency) is determined (red dot). (b) The image is then cropped so the center is at the focal point. (c) The image is then 'foveated' resulting in pixels closer to the center becoming over-represented while pixels close to the edge are under-represented. In this case, the white area on the left of the foveated image is representing the white pixels inside the loop of steel wire of the source image. (d) The result is downsampled uniformly.

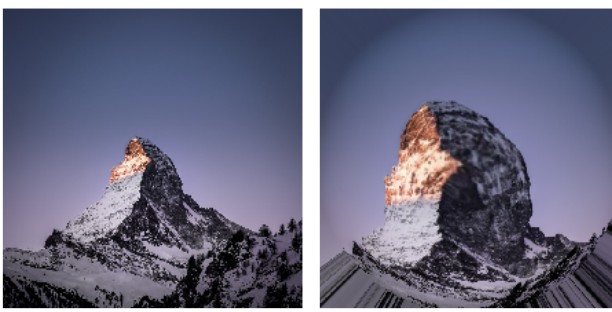

Figure 3: An image before and after cartesian-retinal downsampling. Much like polar foveation, the center of the image is over-represented in the downsample while the extremities are under-represented, proportional to RGC density data.

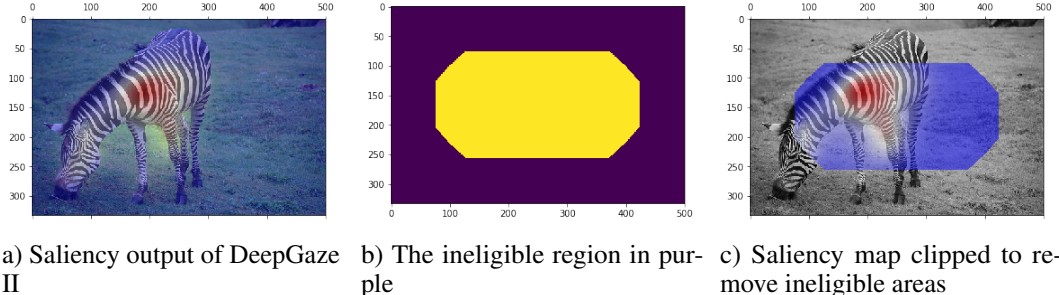

a) Saliency output of DeepGaze II

b) The ineligible region in purple

c) Saliency map clipped to remove ineligible areas

Figure 4: The process of finding a valid saliency map from which the point of highest saliency is chosen. (a) A DeepGaze II model determines a general saliency map. (b) An ineligible region is identified (in purple) where points would result in too much of the resultant crop (20% or more) falling outside the image. (c) The saliency map is clipped and normalized before points are chosen.

## 3 Results

Figure 5 shows training curves for each of the downsampling methods. During training, each crop surrounded one of the three most salient points (with sequential updating of the salience map, as

Table 1: Performance on the validation set

| Model | Top 1 Accuracy | Top 5 Accuracy |
|---|---|---|
| Most Salient: Uniform | 37.66 | 63.03 |
| Most Salient: Polar-Retinal | 33.85 | 57.50 |
| Top-3 Salient: Uniform | 47.72 | 70.95 |
| Top-3 Salient: Polar-Retinal | 47.88 | 70.26 |
| Top-3 Salient: Cartesian-Retinal | 48.95 | 71.79 |
| Top-3 Salient: Multi-Resolution | 47.34 | 69.91 |

described in the Methods) at random. We also separately trained networks with the uniform and polar methods using the single most salient crop. Table 1 summarizes validation performance of the trained models. For Top-3 salient results, predictions were based on three foveations for each image (logits averaged across foveations).

## 4 Discussion

The cartesian method performed best in this study. Each of the foveated methods has a limitation that could potentially be improved in future work. The polar mapping sacrificed translational equivariance (e.g. the same edge detector could respond to a vertical edge at the bottom of the image and a horizontal edge at the side). This might be mitigated by adding rotational equivariance. The cartesian representation wasted pixels at the corners of the image, which limits computational efficiency. Our version of the multi-resolution representation arranged resolutions side-by-side, which introduced edge effects. The resolutions could also be treated as separate input channels. We did not do this because we wanted to hold constant the numbers of parameters and sizes of the representations across models. Given that foveated views seem not to impair object recognition performance, it would be interesting to explore potential benefits within more general vision systems.

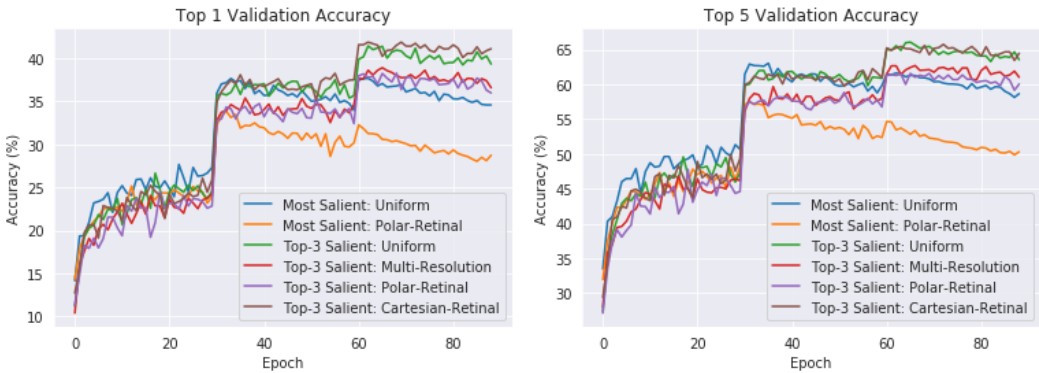

Figure 5: Top 1 (left) and Top 5 (right) validation accuracy during training

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
