# OpenReview forum: "Foveated Downsampling Techniques"
_NeurIPS.cc/2019/Workshop/Neuro_AI — Submitted to Real Neurons & Hidden Units @ NeurIPS 2019_

### Official Review · AnonReviewer1 · 2019-09-25
**An interesting albeit not totally convincing investigation, but the idea is not novel and doesn't work particularly well**

**Clarity:** 4

**Comment:**

This submission investigates the performance of ImageNet classifiers trained on tiny images generated by choosing salient image regions using the DeepGaze II model and applying several different downsampling techniques (uniform, polar-retinal, Cartesian-retinal, and multi-resolution). The authors find that Cartesian-retinal downsampling, which magnifies the central part of the patch, seems to perform marginally better than uniform downsampling in this setting.

Strengths:

The authors investigate several interesting downsampling techniques and describe the results clearly and accurately.

Evaluation is performed on a real dataset (ImageNet) where results can be directly compared to previous results.

Weaknesses:

None of the proposed methods obtain substantial gains over the uniform downsampling baseline, with the best novel method (Cartesian-retinal) achieving an absolute gain of ~1% in top-1 accuracy.

Given the poor performance of all networks relative to baselines that operate at the same image resolution without cropping, I am not convinced that the results would generalize to other settings (or even the same setting with a network architecture better suited to the task).

The idea of polar downsampling is not novel, but previous work (e.g. [1,2]) is not cited.

[1] Elliman, D. G., & Banks, R. N. (1990). Shift invariant neural net for machine vision. IEE Proceedings I (Communications, Speech and Vision), 137(3), 183-187.
[2] Esteves, C., Allen-Blanchette, C., Zhou, X. & Daniilidis, K. (2018). Polar transformer networks. ICLR.

**Category:**

Neuro->AI

**Clarity Comment:**

In general, both the experiments and results are well-described. There were a few things that were unclear to me, but I realize that 4 pages is a rather significant space restriction.

1. Description of training (L34-L36): The authors say "SGD" but the initial DenseNet paper trained with SGD + momentum of 0.9. I'm also curious whether the authors take random crops from the tiny images as is common for ImageNet training or just do random flips.
2. Description of multi-resolution downsampling (L40-L43): The discussion indicates that the multi-resolution representation stacked the images side-by-side, but this was not clear from the description here.
3. It might have been useful to provide examples of input images for all input representations and not merely the polar retinal representation.

**Evaluation:**

2: Poor

**Importance:**

1: Irrelevant

**Importance Comment:**

The setting in which the proposed idea is tested is not fully convincing, and it does not achieve significant gains in this setting. Additionally, the general idea of using a polar transformation followed by a neural network to classify the transformed image is not novel. Overall, I do not believe this paper provides much actionable knowledge, even for those interested in the general idea.

**Intersection:**

4: High

**Intersection Comment:**

The relevance of the structure of the human visual system for machine learning methods is an interesting topic at the intersection of AI and neuroscience. Since natural selection has tuned the structure of the human visual system to be particularly good at processing the visual environment that humans face, it stands to reason that machine learning systems could benefit from adopting aspects of this structure.

**Rigor Comment:**

The authors have tried to design fair experiments, but some details seem a bit problematic. It seems like the authors attempted attempt to control the number of total pixels in each image representation. This is at least a good approximation to the computional cost. The authors train on ImageNet, which is a large-scale dataset well-suited to determining whether the proposed method can be used to improve image classification performance.

What is more problematic is the small size of the images, the selection of the network architecture to process these images, and the poor performance of the baselines relative to previous results operating at the same image size. The authors apply DenseNet-121, which is an ImageNet network, to images of different tiny sizes (32x32, 16x16, and 44x23). This seems weird to do without adjusting the network architecture. DenseNet-121 is intended to operate on ~224x224 pixel images, and downsamples by a factor of 32 throughout the network (by a factor of 4 in the first two layers alone). A CIFAR-10 DenseNet might be more appropriate at this image size. Chrabaszcz et al. [1] show that a network trained on 32x32 pixel ImageNet can achieve 59% top-1 accuracy on the uncropped images. The best 3-crop result in this submission is 49% and the best single-crop result is 38%, so the gap is uncomfortably large. It's difficult to know how to interpret a ~1% gain from the proposed representation in the 3-crop setting given that a >50% relative gain in the 1-crop setting can be obtained simply by changing the architecture.

[1] Chrabaszcz, P., Loshchilov, I., & Hutter, F. (2017). A downsampled variant of ImageNet as an alternative to the CIFAR datasets. arXiv preprint arXiv:1707.08819.

**Technical Rigor:**

2: Marginally convincing

---

### Official Review · AnonReviewer2 · 2019-09-26
**Interesting topic, but implementation choices make it hard to interpret the lack of clear differences among conditions**

**Clarity:** 4

**Comment:**

-- Using an architecture with a native input size that matches the inputs might improve overall accuracies and better reveal differences between downsampling techniques. (Of course this would create a mismatch in input sizes for the Polar vs other methods, but the number of input channels would be almost identical so is perhaps not a major concern)

-- It might be more informative to study the effects of the variable resolution manipulations separately from the effect of severe downsampling. i.e. consider versions of the network with 256x256 (or similar) input sizes, but using each of the three variable-resolution methods.

-- In order to get an impression of how detrimental the various downsampling methods are, it would be helpful to have a baseline case of a network which is trained on the saliency-selected and cropped images, at full resolution (i.e. 256x256, or full network input size). The absolute accuracy values of all networks here are quite low, but some of that is presumably due to the cropping method removing more of the image than standard ILSVRC resizing & cropping approaches would?

-- Description and discussion of results are appropriately measured, and frame the small differences in accuracies in a positive way without over-selling them. Authors identify several drawbacks of their methods and discuss them in an open fashion.

**Category:**

Neuro->AI

**Clarity Comment:**

-- Generally clearly written and well described. Figures are helpful and clearly illustrative of steps in methods.

**Evaluation:**

2: Poor

**Importance:**

2: Marginally important

**Importance Comment:**

-- Identifies an important question, of strong interest both for engineers and for visual neuroscientists. However, the results feel rather preliminary, given that there are many different ways this project could have been implemented, and it's not clear what effects each of the current implementation choices are having.

**Intersection:**

5: Outstanding

**Intersection Comment:**

-- Addresses a strong intersectional question of interest to both fields (how might the variable resolution of mammalian visual sampling affect recognition performance, and could it have computational benefits?)

**Rigor Comment:**

-- The main oddity, as discussed by Reviewer 1, is that, if the hyperparameters of the Densenet are kept at their original values, then this implies that the network was trained on predominantly blank images, with only a small 32x32 pixel central region being the actual input image. Surely this means that the filter sizes and pooling choices within the network were not optimal for the inputs (and perhaps also that training was more difficult, as most input channels didn't contribute to the error gradient)?

-- The accuracies of all networks are quite low compared to usual Imagenet performance of Densenet and similar state of the art models. Within this range of poor performance, there is little difference between the various downsampling methods explored. It would be informative to see a more systematic test of the three factors that could be influencing performance: (1) the saliency-based crop selection, (2) the eight-fold image downsampling, and (3) the three different variable-resolution strategies.

-- The severe eight-fold downsampling is somewhat orthogonal to the question of whether variable resolution can be used to maximise computational resources. It seems like there's a risk that the combination of image cropping and extreme downsampling already reduced the image information to such a point that no meaningful benefits could be obtained by using variable resolution methods. It would be more compelling to see a series of tests with different degrees of downsampling, combined with the different variable resolution methods.

**Technical Rigor:**

2: Marginally convincing

---

### Official Review · AnonReviewer3 · 2019-09-26
**Limited but potentially useful results on foveated downsampling**

**Clarity:** 4

**Comment:**

The authors could perhaps contextualize this work more clearly in either a biological or an engineering motivation: how does their study inform existing theories of foveation, or advance techniques in image recognition? Bandwidth may be an important constraint in future engineered systems, so the finding that foveated downsampling is not a substantial hindrance to image recognition is potentially quite useful.

**Category:**

Neuro->AI

**Clarity Comment:**

The paper is clearly written, and figures are easy to interpret.


**Evaluation:**

2: Poor

**Importance:**

2: Marginally important

**Importance Comment:**

This paper assesses several foveation techniques over a standardized data set and training procedure. The authors find that foveation does not substantially affect image recognition performance, given salience data.

**Intersection:**

3: Medium

**Intersection Comment:**

The paper is mildly motivated by the observation that biological vision exhibits foveation. However, the applications are most relevant to engineered systems, and it is unclear how general the insights will be given the low performance of all of the models.

**Rigor Comment:**

Though the claim of the paper is modest, the methods are explicit and systematic. One issue is that while the techniques all yield similar performance, that performance is not very good.

**Technical Rigor:**

2: Marginally convincing

---

### Decision · Program_Chairs · 2019-10-01

**Decision:**

Reject

**Comment:**

Unfortunately, we had more submissions than we could accept and based on the review process, we have decided not to accept your submission.  Nevertheless, thank you for your submission and interest in our workshop.